



# Operational and experimental snow observation systems in the upper Rofental: data from 2017 - 2020

Michael Warscher[1], Thomas Marke[1], and Ulrich Strasser[1]

[1]Department of Geography, University of Innsbruck, Innsbruck, 6020, Austria

**Correspondence:** Michael Warscher (michael.warscher@uibk.ac.at)

**Abstract.** According to the living data process in ESSD, this publication presents extensions of a comprehensive hydrome-teorological and glaciological data set for several research sites in the Rofental (1891–3772 m a.s.l., Ötztal Alps, Austria). Whereas the original dataset has been published in a first original version in 2018 (https://doi.org/10.5194/essd-10-151-2018), the new time series presented here originate from meteorological and snow-hydrological recordings that have been collected

from 2017 to 2020. Some data sets represent continuations of time series at existing locations, others come from new installa-tions complementing the scientific monitoring infrastructure in the research catchment. Main extensions are a fully equipped automatic weather and snow monitoring station, as well as extensive additional installations to enable continuous observation of snow cover properties. Installed at three high Alpine locations in the catchment, these include automatic measurements of snow depth, snow water equivalent, volumetric solid and liquid water content, snow density, layered snow temperature profiles,

and snow surface temperature. One station is extended by a particular arrangement of two snow depth and water equivalent recording devices to observe and quantify wind-driven snow redistribution. They are installed at nearby wind-exposed and sheltered locations and are complemented by an acoustic-based snow drift sensor.

    The data sets represent a unique time series of high-altitude mountain snow and meteorology observations. We present three years of data for temperature, precipitation, humidity, wind speed, and radiation fluxes from three meteorological stations.

The continuous snow measurements are explored by combined analyses of meteorological and snow data to show typical seasonal snow cover characteristics. The potential of the snow drift observations are demonstrated with examples of measured wind speeds, snow drift rates and redistributed snow amounts in December 2019 when a tragic avalanche accident occurred in the vicinity of the station. All new data sets are provided to the scientific community according to the Creative Commons Attribution License by means of the PANGAEA repository (https://www.pangaea.de/?q=%40ref104365).

# 1  Introduction

Mountain regions are subject to particularly fast environmental changes induced by the rapid development of changing global climate. They are likely to be more vulnerable in the expected consequences on the typical mountain ecosystems (Beniston et al., 2018; Pörtner et al., 2019). Much evidence has been collected and documented in the recent past showing that the rate of climate change induced temperature change can be amplified in high altitudes (elevation-dependent variations), i.e., in the

mountain regions of the world (Ohmura, A., 2012; Pepin et al., 2015; Wang et al., 2018). Recent research focuses on consequent





impacts on precipitation amounts (and rain/snowfall rates), glacial ice losses, changes in snow cover and melt dynamics, and consequent runoff behavior and water supply in mountain regions (e.g., Barnett et al., 2005; Trenberth, K.E., 2011; Beniston et al., 2018; Blöschl et al., 2019). The spatial patterns in the changes in high mountain snow cover, their implications for water storage and release, and their function as protection sheet for glaciers in a changing climate are still poorly understood. E.g.,

Musselmann et al. (2017) and Wu et al. (2018) recently found the counterintuitive effect of slower snowmelt in a warming climate due to the seasonally earlier melting period in the future with less radiative energy input. All these open research questions underline the specific importance of high altitude observations of snow and climate because data in these regions is still sparse compared to low elevations, even in a well exploited mountain range as the European Alps (Matiu et al., 2020).

The Rofental in the Ötztal Alps has developed to a well-recognized high Alpine environmental research basin and coop-

eration platform. Strasser et al. (2018) documented the history of diverse environmental observation data for the Rofental going back to the year of 1850. In this original publication all the available data sets are compiled for the Rofental until 2017. These are archived and accessible at https://doi.pangaea.de/10.1594/PANGAEA.876120. The data documented by Strasser et al. (2018) has been extensively used to investigate snow-hydrological, glaciological and meteorological processes. E.g., Klug et al. (2018) summarized annual mass balances from 2001 to 2011 for the Hintereisferner (HEF) by combining geodetic and

airborne laserscanning data. Rieg et al. (2018) presented the applicability of Pléiades tri-stereo satellite data to derive multi-temporal high-resolution DEMs, and calculated mass balances for the Hochjochferner. Schmieder et al. (2018) used the data for hydrograph separation in a stable water isotope measurement campaign. Glaciological, hydrological, and meteorological data was used by Hanzer et al. (2016, 2018) to validate results of a physically-based hydrological model, and to assess climate change impacts on the hydrology of the catchment. De Gregorio et al. (2019a, b) used data from the catchment to develop

and evaluate novel remote sensing techniques for satellite-based retrieval of snow coverage and SWE. A similar study was conducted by Rastner et al. (2019) who focused on automated mapping of snow cover and snow line altitudes from Landsat data. Zolles et al. (2019) used the data to present an uncertainty assessment of glacier mass and energy balance models. The effect of spatial and temporal flow variations on turbulent heat exchange at the Hintereisferner was investigated by Mott et al. (2020). Stoll et al. (2020) used the data to compare the impact of different model approaches on glacio-hydrological climate

change studies in high-mountain catchments.

Much of the measurement infrastructure documented in (Strasser et al., 2018) has been further maintained without changes. Some of the installations were modified and modernized to meet current technical standards (mostly the ones of operational data transmission). Most important, the existing observation network has been extended with a new automatic weather station, and has been complemented by sensors continuously recording snow cover properties. In the following, we document these

extensions of the observation programme and we present the new data sets that have been recorded between 2017 and 2020. We show the potential of the data and specifically the new snow monitoring sensors by presenting exemplary use cases of data analysis. The new data time series are complementing the continuous meteorological and snow observations in the upper Rofental to support I) improved process understanding of snow and melt dynamics in high mountain regions, II) process model development, evaluation, and application on different scales and for different purposes (regional climate and weather,

glaciology, hydrology, ecology) and III) operational avalanche warning and flood forecasting services.





## 2   The Rofental - site description

The Rofental (98.1 km$^2$, Fig. 1) is a glaciated headwater catchment in the Central Eastern Alps, namely in the upper Ötztal Alps (Tyrol, Austria). It is described in detail in the first data publication by Strasser et al. (2018) (same special issue). Here, we give a short summary of the site description. The Rofental stretches from 1891 m a.s.l. at the gauge at Vent, the lowest
point of the catchment, to 3772 m a.s.l. at the top of Wildspitze, the highest summit of Tyrol. It is characterized by a valley floor in the lower part which is a narrow discontinuous riparian zone typically less than 100 m in width. The average slope of the catchment is 25° and the average elevation is 2930 m a.s.l. (Strasser et al., 2018). The Rofenache is a tributary to the Venter Ache, Ötztaler Ache and the Inn and as such contributing to the Danube system (i.e., the Black Sea). The runoff regime of the Rofenache has not been modified by any measures of hydropower generation and is dominated by the melt of snow and
ice during spring and summer, respectively. The early melt season onset is typically in April. The climate of the Rofental is characterized as an inner Alpine dry type. The mean annual temperature at the station in Vent (1900 m a.s.l., 46.85833° N, 10.91250° E) is 2.5 °C, and total annual precipitation varies between 797 mm in Vent (1982–2003, Kuhn et al. (2006)) and > 1500 mm in the higher altitudes around 3000 m a.s.l. In these higher regions, seasonal snow cover lasts from October until the end of June (Strasser et al., 2018). Maintenance of the monitoring instrumentation and fieldwork in the valley is supported by
the given accessibility - partly also in winter - and logistical infrastructure. A research station at Hintereisferner (HEF, built in 1966 in 3026 m a.s.l.) and one at Vernagtbach (built in 1973 in 2637 m a.s.l.) serve as logistic bases for fieldwork on the two glaciers. Several mountain huts are located in the Rofental, namely the "Vernagthütte/Würzburger Haus" (2755 m a.s.l.), the "Hochjoch-Hospiz" (2413 m a.s.l.), the "Brandenburger Haus" (3277 m a.s.l.) and close by the Austrian-Italian borderline at the Hochjoch the "Schöne Aussicht Schutzhütte" ("Bella Vista", 2845 m a.s.l.), in the Schnalstal glacier ski resort. The
"Rofenhöfe" (2014 m a.s.l.), the highest permanently settled mountain farm in Austria, are well situated as base camp in the lower valley floor, accessible all through the year.

## 3   The observation network and its development

The history of the Rofental catchment as a research basin has been described in detail by Strasser et al. (2018). Apart from its history within UNESCO IHP (http://en.unesco.org/themes/water-security/hydrology), it is a member of several international
research initiatives. The Rofental is a research basin in the framework of the GEWEX INARCH project (http://words.usask.ca/inarch) and part of the ERB Euro-Mediterranean Network of Experimental and Representative Basins (http://erb-network.simdif.com). Further, it is a regular complex site in the LTSER platform Tyrolean Alps (http://lter-austria.at/ta-tyrolean-alps) which belongs to the national and international long term ecological research networks LTER Austria, LTER Europe and ILTER (see e.g. Ohl et al. (2010); Angelstam et al. (2019)). Hintereisferner station is part
of the EU Horizon 2020 INTERACT framework of Arctic (and a few Alpine) research stations (https://eu-interact.org/field-sites/station-hintereis/).

Due to its complex topography, the Rofental is characterized by steep environmental gradients and large spatio-temporal variations of the meteorological conditions. An ongoing effort has been undertaken in the past years to supplement data avail-

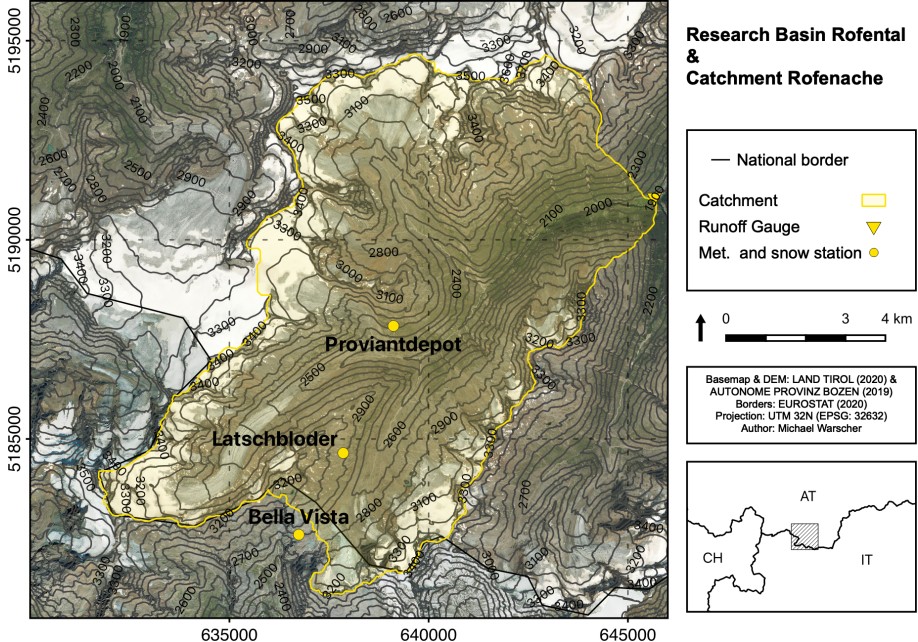

**Figure 1.** The research basin Rofental and the Rofenache catchment (98.1 km$^2$) with the three automatic weather and snow stations Bella Vista, Latschbloder, and Proviantdepot are highlighted. A map displaying the surrounding monitoring stations can be found in Strasser et al. (2018).

able from the lower regions with additional automatic weather station (AWS) installations in the higher elevations. Since the

reported state of the technical instruments in the valley in Strasser et al. (2018), the existing weather stations have been extended and modified at several locations in the catchment. In addition to the standard meteorological variables, multiple snow cover properties are recorded. We here present the extensions and modifications of the meteorological sensors at two AWSs and describe the installation of a new AWS brought into operation in 2019 (Sect. 3.1). These three stations are the locations for the extensive additional snow measurement systems presented in Sect. 3.2.

**3.1 Meteorological stations**

In the uppermost parts of the Hochjoch valley, two AWSs have been brought into operation in 2013 and 2015: Latschbloder (2919 m a.s.l., 46.80106° N, 10.80659° E, Tab. 1, Fig. 1), installed in September 2013, and Bella Vista (2805 m a.s.l., 46.78284° N, 10.79138° E, Tab. 2, Fig. 1), installed in June 2015. The setup and data of these two fully automatic stations have been described in detail by Strasser et al. (2018). In September 2017, the sensors for air temperature, relative humidity, wind speed, and

air pressure at both stations were replaced by new instruments (Tabs. 1 and 2). The Vaisala WXT520 instrument was replaced by E+E E08 sensors for air temperature and relative humidity, by Kroneis 262 (Bella Vista) / Young 05103 (Latschbloder) instruments for wind speed and direction, and by Young 61302V sensors for atmospheric pressure. To complement the station network, a new AWS has been installed in September 2019. The Proviantdepot station (2.737 m a.s.l., Tab. 3, Fig. 1) is an





ensemble of instruments situated on a flat plane built by an old moraine at the orographically left side of the Kesselwandferner.

The slope behind and below the instruments faces south, and from the location of the station one has a panoramic view of the total area of the Hintereisferner. The new instruments are located approximately 20 m east of the totalizing rain gauge that has been in operation since 1952. The centroid of the instrumentations is located at 46.82951° N, 10.82407° E. For all stations, the height of the sensors above ground is at least 1.5 m; in winter, the distance between the snow surface and the sensors can become much smaller, and in extreme snow-rich periods the instruments even can become completely snow-covered. Such

periods can be recognized in the data by typical recordings of zero wind speed and increasing dampening of the other meteorological variables. All three stations Bella Vista, Latschbloder, and Proviantdepot have been equipped with extensive automatic snow cover measurement systems. These systems are presented in Sect. 3.2. The data at all three stations is recorded in 10 min. intervals and transmitted by means of GSM. In Tabs. 1, 2, and 3 the technical sensor specifications are listed in detail.

### 3.2 Automatic snow cover measurements

An important aim in the conceptual development of the Rofental measurement network was the extensive and operational observation of the snow cover and its properties, for which characteristic locations in the high Alpine terrain of the catchment were chosen. Therefore, the three AWS Bella Vista, Latschbloder, and Proviantdepot include extensive automatic measurements of various snow cover properties which are recorded continuously in a 10 min. interval. These comprise observations of snow depth (SD), snow water equivalent (SWE), layered snow temperature profiles, snow surface temperature, liquid and solid

water content of the snowpack, as well as snow drift (see Tab. 1, 2, and 3). In the following, these are named automatic weather and snow stations (AWSS). The data of the AWSSs are operationally used by the European Avalanche Warning Services EAWS and visualized in real-time at https://avalanche.report/weather/measurements and https://www.lawis.at/station/.

### 3.2.1 Bella Vista

The Bella Vista AWSS (2805 m a.s.l., 46.78284° N, 10.79138° E) is located in close vicinity to the "Schöne Aussicht

Schutzhütte". It is located exactly at the central ridge of the Central Eastern Alps, the weather divide between the Northern and Southern Eastern Alps. Generally, the region belongs to the rather dry inner Alpine climate zone. The station is built in small scale heterogeneous terrain at a gentle slope in a barren, rocky landscape. It is affected by wind and frontal systems from both, northern and southern directions. The AWSS has been extended by an additional snow measurement system in October 2019 and September 2020. A Sommer SSG-2 snow scale to measure SWE, an ultrasonic SD sensor, and a snow temperature

profiler have been installed in a depressed location near the main station that is prone to snow accumulation by wind-drift (Fig. 2). The new instrument complements the existing measurements of SWE (by means of a snow pillow), SD (by means of an ultrasonic ranger) and snow temperature profile (by means of a series of temperature sensors at different height levels) located at the main station. The location of the main station is rather exposed and therefore prone to snow erosion by wind. The relation between the exposed and sheltered snow measurements allows for an assessment of the timing and amount of

wind-driven snow redistribution. This technique is illustrated by the data analysis of a blowing snow event that was related to

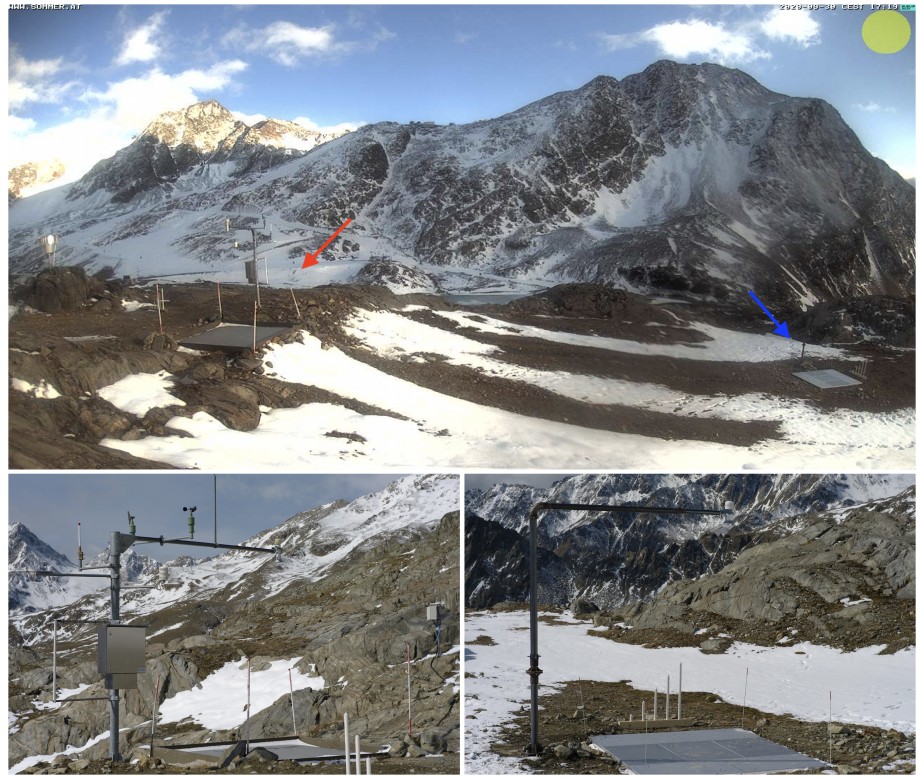

**Figure 2.** Webcam image from the Bella Vista station (top, Sep 30, 2020). The red arrow marks the main AWS and "exposed" snow measurements. The blue arrow marks the additional snow measurements (SD, SWE, and snow temperatures) in the slight depression ("sheltered" location). Close ups of the main AWSS (bottom left) and the secondary snow measurements (bottom right).

the tragic avalanche accident in December 2019 (Sect. 4.2.4). In addition, a new instrument to directly measure the particle flux of drifting snow by means of an acoustic sensor (Sommer SND - Snow Drift Sensor) has been installed in September 2020.

### 3.2.2 Latschbloder

The Latschbloder AWSS (2919 m a.s.l., 46.80106° N, 10.80659° E) is located on a gently sloped plateau below the "Rofen-

bergköpfe" (3229 m a.s.l.) and was chosen as a meteorologically representative measurement for the regional climate that is not largely influenced by steep surrounding slopes and the corresponding local wind systems (Fig. 3). It is located near a totalizing rain gauge that was installed in 1965 and has been equipped with an ultrasonic snow depth sensor (Sommer USH-8) in 2017. In September 2020 an automatic snow temperature profiler that records temperature in the snow cover at the base level and in 20, 40, 60, and 100 cm from the ground has been installed.

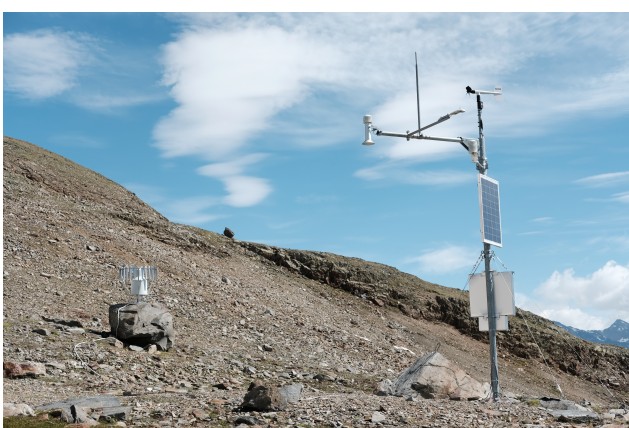

**Figure 3.** The Latschbloder AWSS (2919 m a.s.l.) with an Ott Pluviometer on the left.

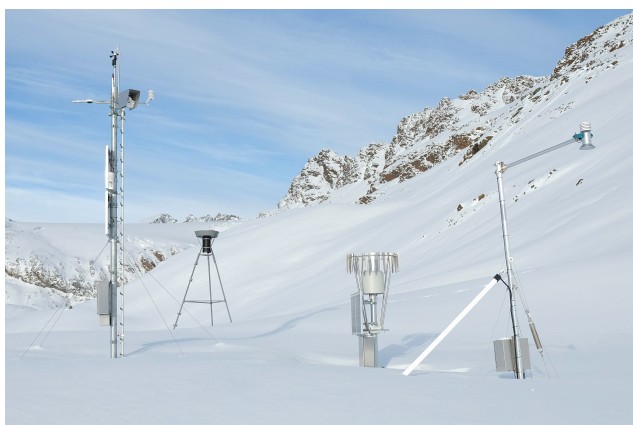

**Figure 4.** The Proviantdepot AWSS (2737 m a.s.l.). The ultrasonic snow depth sensor on the right instrument is part of the Snow Pack Analyzer (SPA). The snow scale is buried right in front of the photographer besides the SPA. There is a second snow depth sensor at the main mast (not visible from this angle). Behind the main mast the old totalizing rain gauge can be seen, in the background the Kesselwandferner.

### 3.2.3 Proviantdepot


The Proviantdepot AWSS (2.737 m a.ssl., 46.82951° N, 10.82407° E, 4) is located on a flat section of a south-facing slope underneath the "Guslarspitzen" (3126 m a.s.l.) halfway between the summit and the "Hochjoch-Hospiz" (2413 m a.s.l.). The station comprises a large set of operational snow cover sensors. SD is measured by a USH-9 ultrasonic device, SWE by means of a SSG-2 snow scale. The temperature of the snow surface is continuously measured by an infrared sensor (Sommer SIR). The layered snow temperatures are recorded analog to the other stations by a SCA temperature profiler. A SPA-2 Snow Pack Analyzer records volumetric contents of solid and liquid water of the snow cover based on measuring the dielectric constants of different frequencies along a flat strap sensor that is spanned within the snowpack.




## 4 The data - new observations and extended data sets

All original research data sets before 2017 have been provided to the scientific community according to the Creative Commons Attribution License by means of the PANGAEA repository (https://doi.org/10.1594/PANGAEA.876120) and offer a comprehensive pool of valuable observations and model forcing data, e.g., for glacier mass balance simulations and mountain catchment hydrology research. They consist of glaciological, hydrological, and meteorological data sets. The new data time series (2017 - 2020) recorded at the AWSSs described in Sect. 3 are available under the same license at the PANGAEA repository (https://www.pangaea.de/?q=%40ref104365).

### 4.1 Meteorological data

Fig. 5 presents the main meteorological variables measured at the three stations Bella Vista, Latschbloder, and Proviantdepot for the years 2017 to 2020. Daily values are shown for air temperature, relative humidity, short-wave radiation, long-wave radiation, and wind speed, as well as monthly values for precipitation. The typical seasonal cycle is visible for temperature and short-wave incoming radiation. Temperature extreme values range from below -20 °C during winter to approximately +10 °C in the summer seasons. Absolute minimum and maximum values in the 10 min data were recorded both at the Bella Vista station with -30.4 °C (February 27, 2018, 3:40am) and +18.8 °C (July 24, 2019, 4:20pm), respectively. Maximum wind speeds were measured during a storm in October 2018 at the Latschbloder station with mean wind speeds of 21 m/s in the 10 min. data (18 m/s hourly, 9 m/s daily), and wind gusts reaching 45 m/s. Short-wave outgoing radiation (Fig. 5) is depending on incoming radiation and strongly controlled by snow coverage on the ground. Values from 200 W/m$^2$ reflected (outgoing) radiation in spring suddenly drop to values as low as 10 W/m$^2$ while incoming radiation is between 200 and 300 W/m$^2$. This occurs in periods when the ground becomes free of snow in spring (May/June) and albedo instantly decreases. The largest precipitation amounts of around 150 to 200 mm/month typically fall during the summer months (June, July, August), whereas during the remaining year, average monthly precipitation totals are approximately 50 mm. For the year 2020, annual precipitation totals are 827 mm at the Bella Vista, 960 mm at the Latschbloder, and 864 mm at the Proviantdepot station. The precipitation values are uncorrected and might - despite the use of heated Pluviometers (Bella Vista and Latschbloder) equipped with wind shelters (all three) - suffer from precipitation gauge undercatch that is typical for high mountain observations with high wind speeds and a large amount of solid precipitation. The two totalizing rain gauges that are located close to the stations recorded long-term annual precipitation totals of 1012 mm (Latschbloder, 1965 - 2016), and 941 mm (Proviantdepot, 1952 - 2016), respectively.

### 4.2 Snow cover data

In the following section, we analyze the first data sets obtained by the various snow observation sensors. Snow depth, snow water equivalent, and snow temperature recordings are operationally used by the avalanche warning and hydrographic services of Tyrol (Austria) and South Tyrol (Italy). The accuracy of the snow drift sensor as well as of the liquid and solid water content measured by the Snow Pack Analyzer are still in discussion. These data are therefore experimental and targeted mainly at research applications.

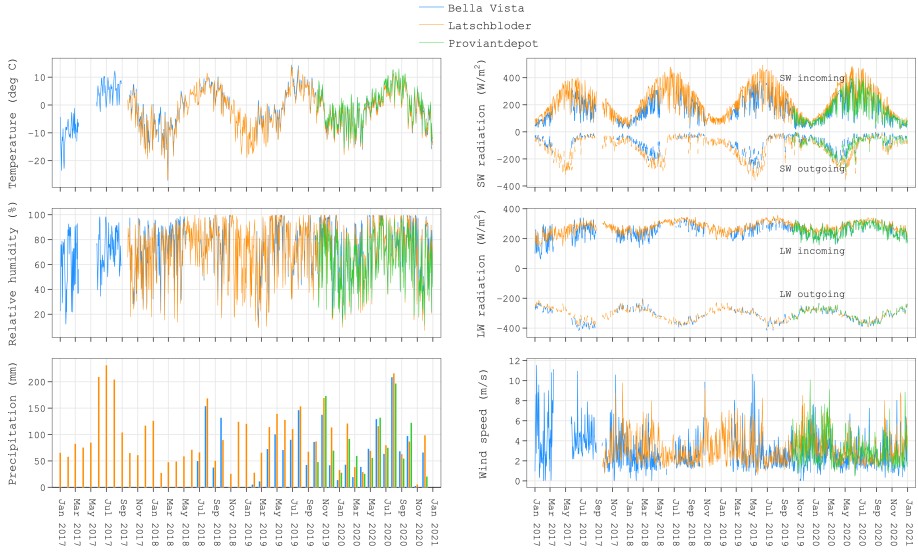

**Figure 5.** Main meteorological variables (daily values) at the three stations Bella Vista, Latschbloder, and Proviantdepot 2017 - 2020. a) air temperature, b) relative humidity, c) precipitation (monthly), d) short-wave radiation, e) long-wave radiation, and f) wind speed.

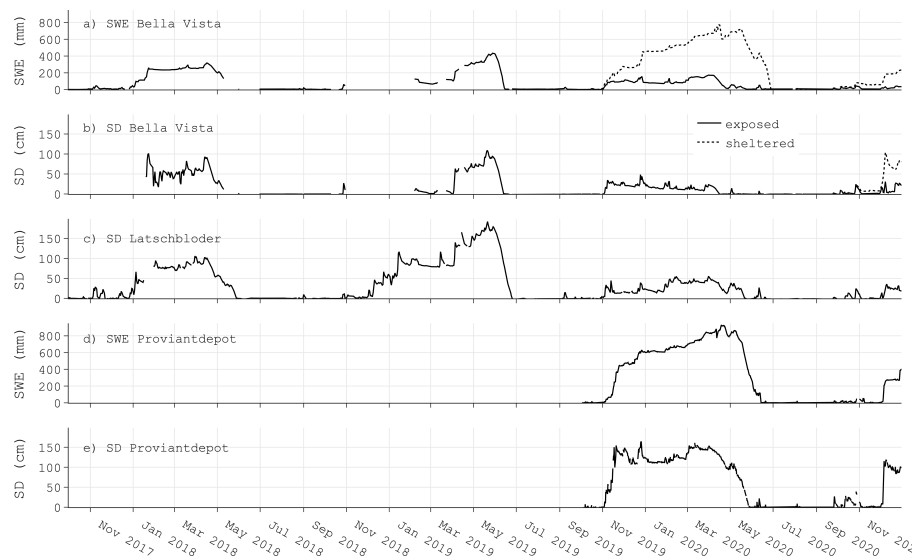

**Figure 6.** Measured SWE and SD (where available) at the Bella Vista double snow station setup (a and b), at the Latschbloder (c) and Proviantdepot (d and e) stations from Oct 1, 2017 to Dec 31, 2020.

### 4.2.1 Snow depth and water equivalent


All available SD and SWE data for the three stations is shown in Fig. 6 (October 2017 to December 2020). In Fig. 7 daily averaged snow depth values for the three stations Bella Vista, Latschbloder, and Proviantdepot are compared for the measuring



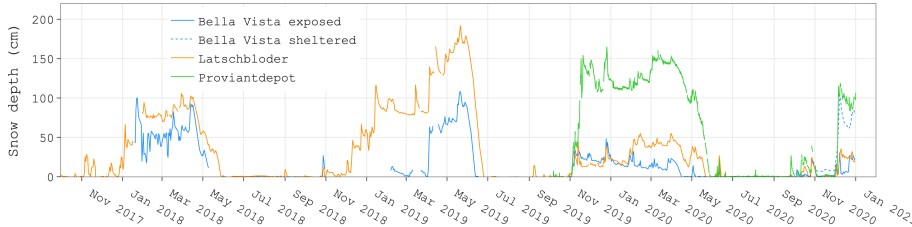

**Figure 7.** Daily averaged snow depth measurements (Oct 2017 - Dec 2020) at the three stations Bella Vista, Latschbloder and Proviantdepot.

period October 2017 to December 2020. Generally winter SD varies between 0.5 and 2 m depending on station and year. At the Bella Vista exposed site usually less SD is measured than at the Latschbloder station. The Bella Vista sheltered site data

is available as of autumn 2020 and already shows large SD values in November and December compared to the exposed site. Maximum SD of close to 2 m were measured at Latschbloder in May 2019. Maximum SD at the Bella Vista exposed site was significantly lower at that time (1 m). The winter 2019/2020 shows large differences in maximum SD between Latschbloder (50 cm) and Proviantdepot (160 cm), indicating that the Proviantdepot received a lot of snowfall from northern frontal systems that did not reach the Latschbloder location further in the south.

**4.2.2   Continuous snow temperature profile**

A snow temperature profile recorded by the Sommer SCA temperature profiler for the whole snow covered period in winter 2019/2020 at the station Proviantdepot is shown in Fig. 8 b). Heights above the ground are 0, 20, 40, 60, 80 and 100 cm, respectively. The corresponding snow depth (Fig. 8 a) shows a typical course over the season. The first large snowfall events in the beginning of November result in a snow depth of 40 cm that grows to 140 cm with more snowfall until the end of the

month. In the following, the snow cover settles slowly and periodically increases again with single snow precipitation events. Snow depth reaches its seasonal peak of 160 cm at the end of December. A long period with only small sporadic snowfall amounts and constant snow settling follows during January and February where SD varies between 110 and 130 cm. Another large snowfall at the end of February leads to a SD of 150 cm. After some settling and little new snow periods, the melting period starts at the beginning of April. Snow melts constantly - only interrupted by some late snowfall at the beginning of

May - until SD is 0 cm at the end of May. The described behaviour and distinguishing melt and settling periods can well be reconstructed using the SWE measurements in Fig. 8 a). SWE steadily increases during snowfall events and stagnates in midwinter when snow is settling but not melting. SWE decreases with the start of the melting period in April. The difference of the time points of SWE and SD reaching zero can be explained by the ultrasonic SD sensor not being directly over the snow scale at the Proviantdepot station (Fig. 4). This is due to constraints in the space around the main mast where the ultrasonic

device is installed. By means of webcam images (not shown here) we identified that there was a wind slab snow patch covering half of the scale while the surrounding was already free of snow. Snow temperature at the base (0 cm) is at 0 °C throughout the whole snow covered period (Fig. 8 b). The elevated temperature sensors at 20, 40, 60, 80, and 100 cm obviously show strong diurnal variations when they are not covered by snow, i.e. measure air temperature at the beginning and end of the snow

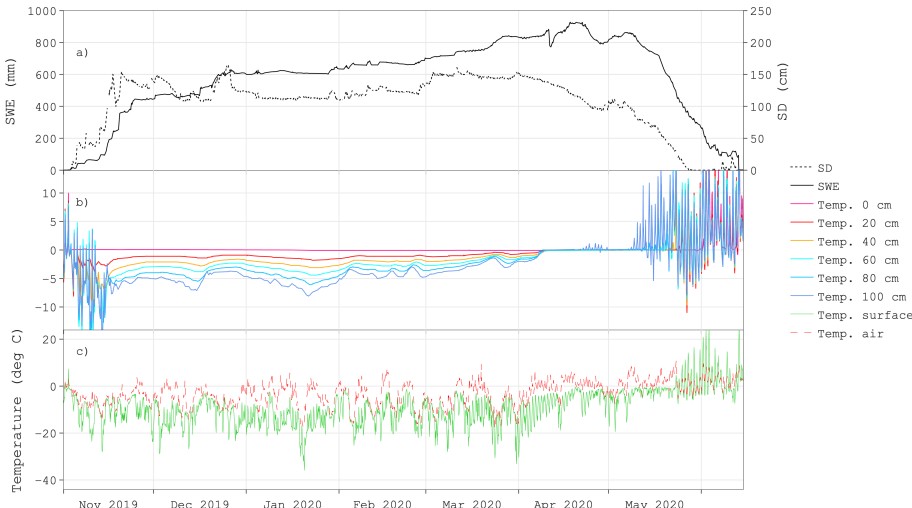

**Figure 8.** Hourly measurements of SD and SWE (a), snow temperatures at the base and in 5 height levels above the ground / in the snow cover (b), and snow surface and air temperature (c) at the Proviantdepot station (Nov 1, 2019 to May 3, 2020).

season. As soon as the sensors are covered by snow, the temperature signal is dampened. A clear snow temperature stratigraphy
with warmer (negative) temperatures in deeper levels (closer to the ground) develops and is retained throughout the season. The layer closest to the surface (100 cm) is influenced the most by prevailing air temperatures and cools down to -8.1 °C on January 21. This cold minimum in the layer approximately 20 cm below the snow surface follows air temperature with a time lag of 1 day. Very cold air temperatures of -17 °C were measured in the night preceding January 20. A minimum snow surface temperature of -36.6 °C was recorded after that cold night at 10:40pm (Fig. 8 c). This time lag is carried on into the
deeper snow layers. This dampening effect can be observed in the data in both directions, i.e. when the snowpack is cooling or warming. The data show a very sharp point in time when the snowpack becomes and stays isotherm, i.e. all layers are at 0 °C (April 9, 2020). Snow starts to melt two days later, i.e. SWE starts to decrease. After the first snowmelt, SWE increases again, but SD steadily decreases, which is a clear indicator for rain falling and percolating into the snow cover. After the small last snowfall at the beginning of May snow melts away steadily.

**4.2.3   Liquid and solid water content (Snow Pack Analyzer)**

Fig. 9 shows measurements of the Snow Pack Analyzer (SPA) at the Proviantdepot station. Due to a logger failure, SPA data is not available before October 2020. Here we show the data from November 25, 2020 to December 31, 2020, i.e. the start of the snow covered period of the winter season 2020/2021. It is characterized by two days of heavy snowfall on December 4 and 5 where SD increases from 10 to 120 cm (Fig. 9 d). Thereafter that the snowpack settles slowly with a next significant
snowfall at the end of December. Fig. 9 a) shows measured solid and liquid volumetric water content measured by the SPA, and the corresponding snow density measurement in Fig. 9 b). It is clearly shown that ice content slowly increases with settling of the new snow, and therefore, snow density increases from 100 kg/m$^3$ to 260 kg/m$^3$ in late December. Liquid water content





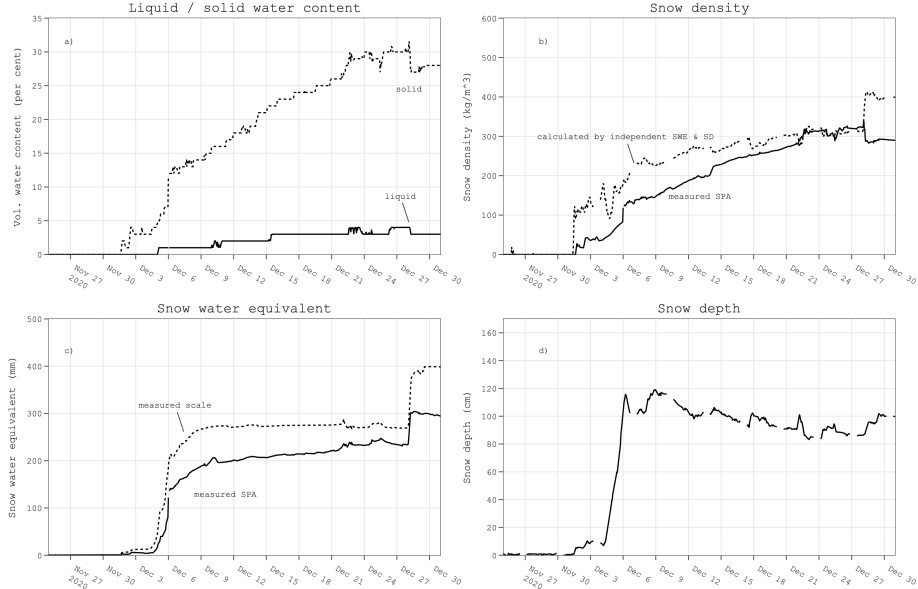

**Figure 9.** Liquid and solid water content measured by SPA (a), snow density as measured by SPA and calculated from independent measurements of SWE and SD (b), SWE measured by SPA and by snow scale (c), and measured SD by ultrasonic device. All data: hourly averages at the Proviantdepot station from Nov 25, 2020 to Dec 31, 2020.

remains low at approximately 2.5 % during that cold period. For comparison, Fig. 9 b) shows measured snow density (SPA) to snow density that was calculated by measured SWE (snow scale) and SD (independent ultrasonic sensor). These snow density

values show good agreement in terms of absolute values and variations with densification processes. However, there is a distinct deviation at the end of December where SPA snow density decreases while the independently measured density increases. As SD increases at that point, the SPA values seem more plausible. The decreasing SPA volumetric liquid and solid water content during that snowfall correspond to the decrease in snow density. Measured air temperatures are in the range of -10 °C during that snowfall (see Fig. 5) making it very unlikely that there is very wet and dense new snow that would be more dense than the

existing snow cover. The increase in snow density in the calculated values can be explained by SWE measurements increasing unproportionally to the SD increase. This can only be explained by the slight difference in measurement location of SD and SWE described above and in the caption of Fig. 4. Please note that there might be additional deviations caused by the slightly different locations of the measurement systems. However, apart from the offset caused by the location (i.e. more snow at the snow scale point), the snow scale SWE is generally in good agreement with the SPA SWE (Fig. 9 c).

**4.2.4   Snow drift measurements**

In Fig. 11, a snow drift event is analysed based on the data recorded by the described instrumentation. This includes data from an acoustic based snow drift sensor (Fig. 10, Sommer SND). The system continuously records snow particle flux in the air by registering the change in acoustic pressure caused by bypassing particles colliding with the cylindrical sensor tube.



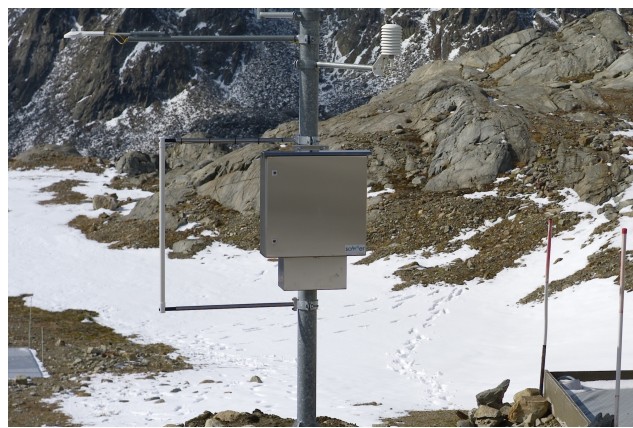

**Figure 10.** The Sommer SND snow drift sensor installation at the Bella Vista AWSS. The sensor elements are placed in the cylindrical tube that is vertically held in place by two mounting arms building a rectangle shape (left of the logger box). In the background (lower left in the photo), the snow scale in the slight depression is visible (sheltered site, sink for snow deposition). The snow pillow besides the main AWSS mast (lower right in the picture) measures SWE at the exposed location.

The measurement principle was first described by Chritin et al. (1999). The accuracy of different versions of the sensor in

quantifying snow flux was discussed in literature by, e.g., Jaedicke, C. (2001); Lehning et al. (2002); Cierco et al. (2007); Trouvilliez et al. (2015) with differing results. However, it is still the only way to continuously measure and detect drifting snow events with a certain reliability (He, S. and Ohara, N., 2017). Cumulated measured snow flux per hour is shown in Fig. 11 d) during a blowing snow event at the Bella Vista station in the night of December 4, 2020. Fig. 11 also includes time series of hourly SD, SWE, mean and maximum wind speeds, as well as precipitation and temperature from December 3 to 5, 2020.

At December 5, 7pm measured mean wind speeds increase from 2 to 8 m/s. Wind gusts range up to 22 m/s during the storm that lasts the night and ends in the morning the next day (6am). Temperatures are very low during the period varying from -7 to -12 °C. The storm is accompanied by snow precipitation in the range of 0.5 to 1.7 mm/h. When comparing SD and SWE at the exposed and sheltered site, erosion and deposition of snow can clearly be identified. SWE at the exposed location decreases from December 4, 7pm to 11pm despite incoming snow fall. The time frame exactly matches the high wind speed recordings.

Contrarily, SWE at the sheltered locations continuously increases as do the corresponding SD values. In contrast, SD at the exposed site increases with snowfall while wind speed is still low and strongly decreases with high wind-speeds shortly after (December 4, 7pm) from 20 to almost 0 cm. In the same timeframe, SD at the sheltered site increases from 20 to 40 cm. These changes are illustrated by webcam images before and after the blowing snow event in Fig. 12.

The snow drift sensor detects particle mass fluxes beginning on December 4, 4pm and shows a strong signal when wind

speeds are high. The lower rates until 7pm suggest that the measured flux is produced by blowing snowfall that has not yet reached the ground as wind speeds are still low, e.g gusts of 5 m/s at 6pm. This is confirmed as both SD values stay constant or increase in that period. SD at the exposed site decreases strongly from 7 pm to 11pm. These are also the times with highest wind speeds and largest measured drift by the SND, i.e. in addition to blowing snowfall, wind speed exceeds the snow erosion



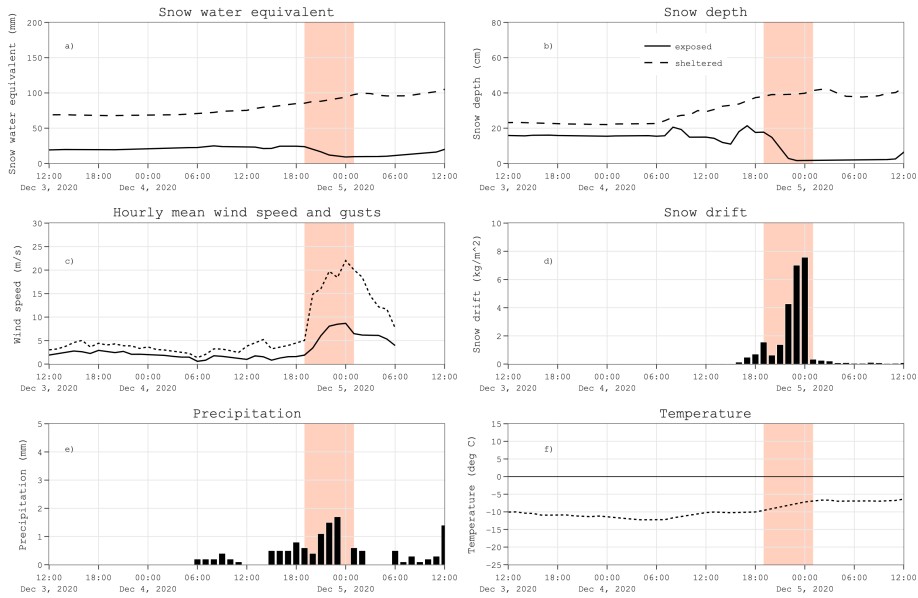

**Figure 11.** Measurement of a snow drift event at the Bella Vista station. Data from Dec 3, 2020 (12:00pm) to Dec 5, 2020 (12:00pm): hourly data of SWE (a) and SD (b) each at the exposed and sheltered point, snow drift (c), precipitation (d), and air temperature (e). The period of the main blowing snow event is highlighted in red.

threshold. To analyze the snow mass flux in the air and at the two ground measurements, we calculated the changes in SD and
SWE, as well as cumulated precipitation and snow particle drift flux from December 4, 7pm to December 5, 1am, i.e. the period
with the strongest signals (Tab. 4). We also show calculated equivalents of SD and SWE for all measurements assuming a snow
density of 100 kg/m$^3$ (new snow) to compare the mass fluxes and storage changes. The measured snow particle flux of 22.8
kg/m$^2$ (equivalent of 22.8 cm SD or 22.8 mm SWE with the assumed density of 100 kg/m$^3$) in total well corresponds to the
changes of -16.1 mm / +14.1 mm SWE (exposed/sheltered point) added the 5.9 mm of snowfall hitting the instrument before
reaching the ground. However, any snow drift values are measurements of a horizontal flux through an area in the atmosphere
of 1 m$^2$ which can not be directly compared to snow depth or SWE changes per m$^2$ at point A or B on the ground. For a full
mass balance of such a complex process, many more variables have to be taken into account (e.g., shear stress and strength,
erosion threshold, suspension time, travel speed and distance, erosion and deposition zones). However, the instrument is able
to detect the snow drift event that can also be observed by SD and SWE measurements and give an estimate about the general
scale of the event. When a longer time series will be available, we will investigate further potential of an automatic detection
and assessment of magnitude of blowing snow events and a possible statistical relation to describe changes in SD / SWE at the
two measurement locations in dependence of measured snow drift values. These results obviously are neither a full assessment
of the sensor accuracy, nor an exact measurement of transported snow mass from point A to point B, but rather a showcase of
the potential of the sensor and the double station setup to a) detect (avalanche / snow slab-critical) snow drift events, and to
b) estimate the amounts of wind-driven snow redistribution in the surrounding terrain. Lehning et al. (2002) pointed out the

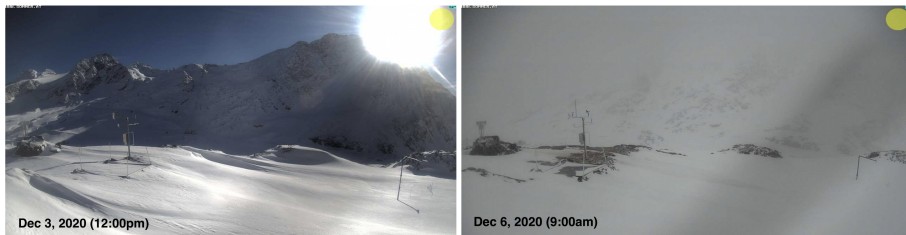

**Figure 12.** Webcam images before (left, Dec 3, 12:00pm) and shortly after the blowing snow event (right, Dec 6, 2020, 9:00am). The main AWSS (exposed location) can be identified in the left side of the pictures, the complementing, second SWE and SD measurements are located at the mast on the right side in the pictures (sheltered location).

potential of this semi-quantitative approach for avalanche warning applications. The accuracy of the sensor in quantifying snow transport rates and its relation to redistributed snow on the ground and at the SD and SWE measurement points will further be investigated when a longer time series is available. Additional field campaigns will be carried out using a mobile terrestrial laser scanning device to measure the spatial distribution of snow depth in the small-scale heterogeneous terrain around the
AWSS.

In the following, we present an additional example of data analysis in the winter season 2019/2020 where an avalanche accident happened in the region. On December 28, 2019 a large avalanche buried several skiers in the ski resort at the Val Senales Glacier, South Tyrol. During the tragic accident three skiers were fatally injured. On the day of the accident, the avalanche report assigned a "considerable" level of danger for the area. The main source of danger was identified to be wind-
blown snow. Strong northerly winds redistributed fresh snow from the previous day to form wind slabs that poorly bonded to the old snowpack. The scene of the accident is located in close proximity to the station Bella Vista (Fig. 13). The instrument setup described above allowed for an assessment of meteorological conditions and wind-driven snow redistribution preceding the avalanche.

Fig. 14 shows wind speed and snow conditions preceding the avalanche. The winter season began with a series of snowfall
events in November. Snow depth increased from 0 to 40 cm from November 3 to November 10 at the exposed location of the Bella Vista station (Fig. 14 a). Snow depth and SWE at the exposed location shows only little variation until December 22 with only small amounts of snowfall in between. However, SWE at the sheltered site shows strong increases during that period resulting in 260 mm SWE compared to less than 100 mm SWE at the exposed location at the beginning of December. This offset between the nearby measurements grows even larger in the days preceding the time of the avalanche at the end of
December (450 mm SWE to 100/150 mm SWE) and corresponds to 400% larger snow masses at the sheltered site compared to the exposed location. The horizontal distance between the two devices is only about 30 m. Fig. 14 c) shows daily mean wind speed and gusts during the corresponding period. It is well discernable that the increase in SWE at the sheltered site occurs when high mean wind speeds including strong gusts prevail. When looking closely at single events, e.g. the time before the avalanche (Fig. 14 b and d), transport of snow from wind-exposed to -sheltered locations can be observed. On December 27 in
the afternoon, mean wind speed began to strongly increase to 8 m/s with gusts reaching 18 m/s and decreased again at 00:00.

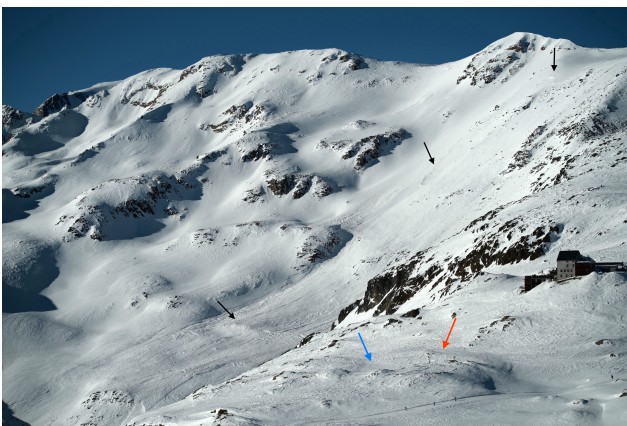

**Figure 13.** Photograph showing the slope of the Dec 28, 2019 avalanche in the background and the location of the Bella Vista station 4 days after the accident. The red arrow marks the Bella Vista AWSS with the "exposed" SD and SWE sensors, and the snow drift sensor, the blue arrow points at the location of the "sheltered" SWE snow scale. The black arrows show from top to bottom the main tear-off edge, secondary edges and avalanche debris. (Image: Peter Höller, Jan 01, 2020)

The wind-blown snow transport can well be identified in the measured data. SWE at the exposed site decreases by 100 mm whereas SWE at the sheltered site strongly increases by 130 mm during this period. The avalanche was triggered at 12:12pm the next day.

## 5 Conclusions

We presented the recent developments in the Rofental research catchment as an extending update to the overview of the existing long time series for the region as documented in Strasser et al. (2018). The three AWSSs of which two were extended and one has been newly deployed, and in particular the various continuous snow cover measurements were described in all technical details.

First and foremost, an extensive observational data set was presented and documented. The data has the potential to be 325 used in different scientific fields, as well as in operational applications. Especially the combination of records of the standard variables of an AWS with the various snow measurements at different high Alpine locations in addition to the rich other glaciological, hydrological, and meteorological data documented in Strasser et al. (2018) is a very seldomly available data set. The presented data can be of use for tackling various research questions in the context of coupled climate and glacier evolution, snow and glacier hydrology, water resources in mountainous regions, or model development and evaluation. The continuous, 330 automatic meteorological and snow observations are already used operationally for assessing regional avalanche danger and forecasting potential flood periods. Secondly, we presented insights into fairly new sensor technologies and an innovative setup of proven instruments to quantify snow redistribution. The data might be used, e.g., for automatic operational detection of



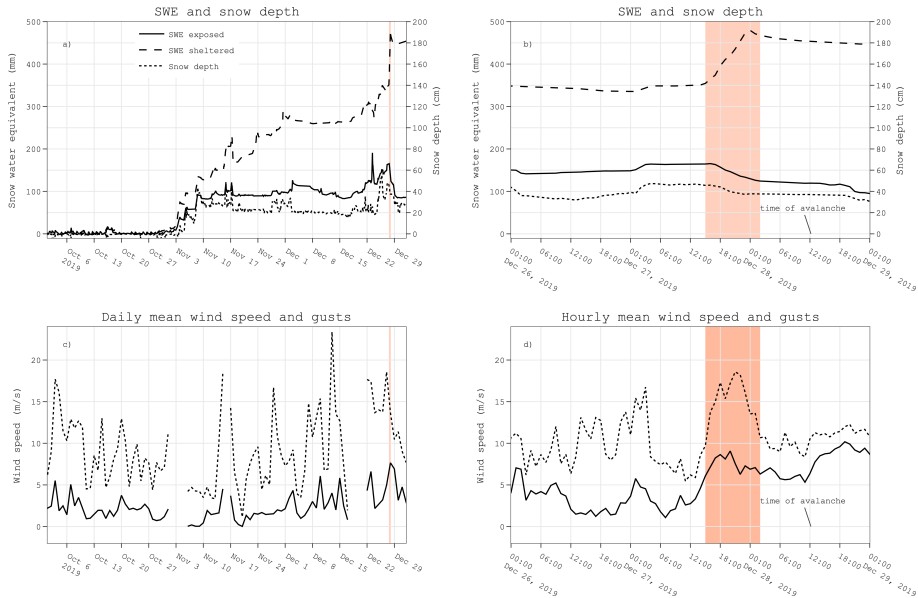

**Figure 14.** Measurements of snow cover and wind speed at the Bella Vista station in the period of Oct to Dec 2019 that lead to an avalanche accident on Dec 28 (12:12pm). a) SWE for wind-exposed and -sheltered location, snow depth for sheltered location; b) same for Dec 26 to 28, c) daily mean wind speed and gusts; d) same for Dec 26 to 28. The distinct wind-driven snow redistribution event is highlighted in red.

avalanche-critical blowing snow situations to support avalanche warning services. Finally, the recordings of the snow sensors were used to draw pictures of several important seasonal snow processes at the event-scale in the region.

All meteorological and snow monitoring installations in the Rofental are intended to stay in operation for the long-term. We will continue to provide the recordings to the community via the Pangaea repository, and elaborate on scientific publications documenting the data, showing examples of important snow processes and events, and collect citations to scientific publication which make use of the data.

## 6   Data availability

The data sets presented here are all available under the CC BY 4.0 license at PANGAEA (https://www.pangaea.de/?q=%40ref104365). The data extends the previous Rofental data collection which is also hosted under the same license at PANGAEA (https://doi.org/10.1594/PANGAEA.876120).

*Author contributions.*   US, TM, and MW designed the station network and sensor concepts and conducted the installations. MW analysed the data and prepared the manuscript with contributions from all co-authors.



*Competing interests.* The authors declare that no competing interests are present.

*Acknowledgements.* We thank Erwin Rottler and Rainer Prinz for support in field work, Carsten Becker for support in data processing, Peter Höller for providing the photograph of the avalanche slope, and Paul Grüner and his team from the "Schöne Aussicht Schutzhütte" for providing comfort and very valuable support during station installation and maintenance work. We also thank Amelie Driemel for the support in data processing and thankfully acknowledge the PANGAEA service for hosting the data. The LTSER platform Tyrolean Alps belongs to the

national and international long term ecological research network (LTER-Austria, LTER Europe and ILTER). The infrastructure is financially supported by the University of Innsbruck, Faculty of Geo- and Atmospheric Sciences and is part of its Research Area "Mountain Regions".





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



**Table 1.** Climate and snow variables recorded by the sensors installed at the station Latschbloder (2919 m a.s.l., 46.80106° N, 10.80659° E). Accuracy according to technical data sheets of the manufacturers. Original temporal resolution of the data records is 10 min.

| Variable | Sensor | Period of operation | Resolution and accuracy | Unit |
|---|---|---|---|---|
| Air temperature | E+E E08 | Since Sep 2017 | < 0.5 °C[1] | °C |
| | Vaisala WXT520 | Sep 2013 to Sep 2017 | 0.1 °C ± 0.3 °C | |
| Relative humidity | E+E E08 | Since Sep 2017 | ± 2 % RH (0–90 % RH), ± 3 % RH (90–100 % RH) | % |
| | Vaisala WXT520 | Sep 2013 to Sep 2017 | 0.1 % ± 3 % RH (0–90 % RH), 0.1 % ± 5 % RH (90–100 % RH) | % |
| Precipitation | Ott Pluvio 2 v. 200 (unheated) with wind shelter | Since Jul 2014 | 0.01 mmh$^{-1}$ ± 1 % | mm |
| | Friedmann tipping bucket | Sep 2013 to Jun 2014 | - | mm |
| | Vaisala WXT520 | Sep 2013 to Sep 2017 | 0.01 mmh$^{-1}$ ± 5 % | mm |
| Wind speed and direction | Young 05103 | Since Sep 2017 | ± 0.3 ms$^{-1}$ (speed), ± 3 ° (direction) | ms$^{-1}$ and ° |
| | Vaisala WXT520 | Sep 2013 to Sep 2017 | 0.1 ms$^{-1}$ ± 3 % (speed) 1°± 3 % for 10 ms$^{-1}$ (direction) | ms$^{-1}$ and ° |
| Wind gust and direction | Young 05103 | Since Sep 2017 | ± 0.3 ms$^{-1}$ (speed), ± 3 ° (direction) | ms$^{-1}$ and ° |
| | Vaisala WXT520 (gust speed only) | Sep 2013 to Sep 2017 | 0.1 ms$^{-1}$ ± 3 % (speed) | ms$^{-1}$ |
| Radiative fluxes (short- and longwave) | Kipp & Zonen CNR 4 | Since Sep 2013 | 10-20 Wm$^{-2}$ (incoming) 5-15 Wm$^{-2}$ (outgoing) | Wm$^{-2}$ Wm$^{-2}$ |
| Atmospheric pressure | Young 61302V | Since Sep 2017 | 0.2 hPa (25 °C), 0.3 hPa (-40 to 60 °C) | hPa |
| | Vaisala WXT520 | Sep 2013 to Sep 2017 | 0.1 hPa ± 0.5 hPa (0 to 30 °C) 0.1 hPa ± 1.0 hPa (-52 to 60 °C) | hPa |
| Snow depth | Sommer USH-8 | Since Sep 2017 | 1 mm ± 0.1 % | mm |
| Snow temperature profile (0, 20, 40, 60, 80, 100 cm) | Sommer SCA snow temperature profile sensor | Since Sep 2020 | ± 0.3 °C | °C |

[1]depending on air temperature; see technical data sheet of the manufacturer.





**Table 2.** Climate and snow variables recorded by the sensors installed at the station Bella Vista (2805 m a.s.l., 46.78284° N, 10.79138° E). Accuracy according to technical data sheets of the manufacturers. Original temporal resolution of the data records is 10 min.

| Variable | Sensor | Period of operation | Resolution and accuracy | Unit |
|---|---|---|---|---|
| Air temperature | E+E E08 | Since Jul 2015 | < 0.5 °C[1] | °C |
| | Vaisala WXT520 | Jul 2015 to Sep 2017 | 0.1 °C ± 0.3 °C | |
| Relative humidity | E+E E08 | Since Jul 2015 | ± 2 % RH (0–90 % RH), ± 3 % RH (90–100 % RH) | % |
| | Vaisala WXT520 | Jul 2015 to Sep 2017 | 0.1 % ± 3 % RH (0–90 % RH), 0.1 % ± 5 % RH (90–100 % RH) | % |
| Precipitation | Ott Pluvio 2 v. 200 (heated) with wind shelter | Since Jul 2015 | 0.01 mmh$^{-1}$ ± 1 % | mm |
| | Vaisala WXT520 | Jul 2015 to Sep 2017 | 0.01 mmh$^{-1}$ ± 5 % | mm |
| Wind speed and direction | Kroneis 262 | Since Jul 2015 | ± 0.2 ms$^{-1}$ (speed) | ms$^{-1}$ and ° |
| | Vaisala WXT520 | Jul 2015 to Sep 2017 | 0.1 ms$^{-1}$ ± 3 % (speed) 1°± 3 % for 10 ms$^{-1}$ (direction) | ms$^{-1}$ and ° |
| Wind gust and direction | Kroneis 262 | Since Jul 2015 (direction since Sep 2017) | ± 0.3 ms$^{-1}$ (speed) | ms$^{-1}$ and ° |
| | Vaisala WXT520 (gust speed only) | Jul 2015 to Sep 2017 | 0.1 ms$^{-1}$ ± 3 % (speed) | ms$^{-1}$ |
| Radiative fluxes (short- and longwave) | Kipp & Zonen CNR 4 | Since Jul 2015 | 10-20 Wm$^{-2}$ (incoming) 5-15 Wm$^{-2}$ (outgoing) | Wm$^{-2}$ Wm$^{-2}$ |
| Atmospheric pressure | Young 61302V | Since Sep 2017 | 0.2 hPa (25 °C), 0.3 hPa (-40 to 60 °C) | hPa |
| | Vaisala WXT520 | Jul 2015 to Sep 2017 | 0.1 hPa ± 0.5 hPa (0 to 30 °C) 0.1 hPa ± 1.0 hPa (-52 to 60 °C) | hPa |
| Snow depth (exposed loc.) | Sommer USH-8 | Since Sep 2017 | 1 mm ± 0.1 % | mm |
| Snow depth (sheltered loc.) | Sommer USH-9 | Since Sep 2020 | 1 mm ± 0.1 % | mm |
| Snow water equivalent (exposed loc.) | Sommer snow pillow 3x3 | Since Sep 2017 | unknown | mm |
| Snow water equivalent (sheltered loc.) | Sommer SSG-2 snow scale | Since Oct 2019 | 0.1 mm ± 0.3 % | mm |
| Snow temperature profile (0, 20, 40, 60, 80, 100 cm, exposed loc.) | Sommer SCA snow temperature profile sensor | Since Sep 2017 | ± 0.3 °C | °C |
| Snow temperature profile (0, 20, 40, 60, 80, 100 cm, sheltered loc.) | Sommer SCA snow temperature profile sensor | Since Sep 2017 | ± 0.3 °C | °C |
| Snow drift | Sommer SND snow drift sensor | Since Sep 2020 | unknown | gm$^{-2}$ |

[1]depending on air temperature; see technical data sheet of the manufacturer.





**Table 3.** Climate and snow variables recorded by the sensors installed at the station Proviantdepot (2737 m a.s.l., 46.82951° N, 10.82407° E). Accuracy according to technical data sheets of the manufacturers. Original temporal resolution of the data records is 10 min.

| Variable | Sensor | Period of operation | Resolution and accuracy | Unit |
|---|---|---|---|---|
| Air temperature | E+E E08 | Since Oct 2019 | < 0.5 °C[1] | °C |
| Relative humidity | E+E E08 | Since Oct 2019 | ± 2 % RH (0–90 % RH), ± 3 % RH (90–100 % RH) | % |
| Precipitation | Ott Pluvio 2 v. 200 (heated) with wind shelter | Since Oct 2019 | 0.01 mmh$^{-1}$ ± 1 % | mm |
| Wind speed and direction | Young 05103 | Since Oct 2019 | ± 0.3 ms$^{-1}$ (speed), ± 3 ° (direction) | ms$^{-1}$ and ° |
| Wind gust and direction | Young 05103 | Since Oct 2019 | ± 0.3 ms$^{-1}$ (speed), ± 3 ° (direction) | ms$^{-1}$ and ° |
| Radiative fluxes (short- and longwave) | Kipp & Zonen CNR 4 | Since Oct 2019 | 10-20 Wm$^{-2}$ (incoming) 5-15 Wm$^{-2}$ (outgoing) | Wm$^{-2}$ Wm$^{-2}$ |
| Atmospheric pressure | Young 61302V | Since Oct 2019 | 0.2 hPa (25 °C), 0.3 hPa (-40 to 60 °C) | hPa |
| Surface temperature | Sommer SIR surface temperature sensor | Since Oct 2019 | unknown | °C |
| Snow depth | Sommer USH-9 | Since Oct 2019 | 1 mm ± 0.1 % | mm |
| Snow water equivalent | Sommer SSG-2 snow scale | Since Oct 2019 | 0.1 mm ± 0.3 % | mm |
| Snow temperature profile (0, 20, 40, 60, 80, 100 cm) | Sommer SCA snow temperature profile sensor | Since Oct 2019 | ± 0.3 °C | °C |
| Snow density | Sommer SPA-2 snow pack analyzer | Since Sep 2019 | unknown | kgm$^{-3}$ |
| Snow liquid water content | Sommer SPA-2 snow pack analyzer | Since Sep 2019 | unknown | vol % |
| Snow ice content | Sommer SPA-2 snow pack analyzer | Since Sep 2019 | unknown | vol % |

[1]depending on air temperature; see technical data sheet of the manufacturer.

**Table 4.** Cumulated measured snow mass fluxes and storage changes from Dec 4, 7pm to Dec 5, 1am, 2020 and calculated SD and SWE equivalents with assumed snow density of 100 kg/m$^3$.

| | Precipitation | SD exposed | SD sheltered | SWE exposed | SWE sheltered | Snow drift |
|---|---|---|---|---|---|---|
| Measured | 5.9 mm | -16.1 cm | +3.4 cm | -14.1 mm | +12.2 mm | 22.8 kg/m$^2$ |
| SD equivalent (assumed density 100 kg/m$^3$) | *+5.9 cm* | -16.1 cm | +3.4 cm | -14.1 cm | *+12.2 cm* | *22.8 cm* |
| SWE equivalent (assumed density 100 kg/m$^3$) | +5.9 mm | *-16.1 mm* | *+3.4 mm* | -14.1 mm | +12.2 mm | 22.8 mm |