# Peer review of "Operational and experimental snow observation systems in the upper Rofental: data from 2017 - 2020"

_Earth System Science Data, 2021_

## Author Comment (AC1)

**Reply to RC1 (Author Comment on essd-2021-68)**

Thank you very much for the very valuable review of our manuscript. We address the questions, issues, suggestions, and remarks in the following point by point (response in blue).

Anonymous Referee #1

Referee comment on "Operational and experimental snow observation systems in the upper Rofental: data from 2017–2020" by Michael Warscher et al., Earth Syst. Sci. Data Discuss., https://doi.org/10.5194/essd-2021-68-RC1, 2021

Review

The current manuscript draft on « Operational and experimental snow observation systems in the upper Rofental: data from 2017 - 2020" describes new weather and snow sensors installed and corresponding data collected in the Rofental catchment.

General comments:

Unfortunately, with the exception of the snow drift sensor, the authors miss the opportunity to introduce the new sensors in detail and to describe the applied data curation.

We added more technical specifications of the other new sensors, specifically the SPA, as this sensor is not as commonly used as the others. The unique double station setup at Bella Vista is extensively explained together with the snow drift sensor.

Moreover, the available data are sometimes carelessly interpreted without any critical plausibility assessment or references to other studies. The possibility of wrong measurements, especially regarding SPA and SGG, was not considered. If really no manual control measurements were performed during the three years, it should at least be mentioned and explained.

We significantly improved our data analyses in two cases, including the clarification of an issue that was brought up by Reviewer 2. For this we included an explanation for the source of error (snow bridging of the SSG). We now point to potential measurement errors and added a respective discussion (see below). On top of this we will add a respective remark to the respective PANGAEA data set.

Nevertheless, the new measurement infrastructure and the corresponding data are worth to be published as soon as the following points have been addressed:

- An overview is missing about what has already been documented in Paper I and what is now newly documented in this paper. Has anything been abandoned?

  We added an explanation in the revised manuscript. Besides the technical updates at the three stations presented here, the observation network exactly operates in the state described by Strasser et al. 2018. No monitoring devices or stations have been abandoned except the documented replacements.

- There is no information about any quality assurance or quality control procedures applied to the data. There is no information about the frequency data are downloaded and screened, if at all?

  *All raw data are transferred via GSM network every 10 minutes to operational servers with automated backups. Screening is manual, but continuous. From the server, the data is annually downloaded, quality checked and corrected and then uploaded to the PANGAEA open access repository. We decided to perform only basic error filtering. We updated the manuscript accordingly and we will add an additional remark to the PANGAEA repository for the respective data sets.*

- The snow drift sensor is explained and referenced in detail. In contrast, e.g. information and literature about the SPA and its measured quantities is missing entirely. For example, the difference in data series S1 and S2 listed in the data set is not explained at all.

  *We added the lacking information about the data series S1 and S2, as well as existing literature on the device and a respective discussion about the SPA.*

- Several times snow fall or snow accumulation is mentioned (e.g. L198-199) without including information of the concurrent precip data. For example, the case mentioned in L198-199 is contradicting the precip signal!

  *While the signal is not contradicting (precipitation at Latschbloder is higher in Dec. 2019 and Feb. 2020 and more or less equal to the one at Proviantdepot in Nov. 2019 and Jan. 2020), we agree that this cannot explain the difference in SD alone, particularly in Nov. 2019. We also checked temperatures during the period (rain/snow) but it is well below 0 °C for both stations. We removed this misleading interpretation from the manuscript.*

  Additionally, the case in L228-229 can't be true because the clearly negative temperatures demonstrate that the reason for the SWE increase can't be rain!

  *While the temperatures are not constantly negative during the period we refer to, there is very little precipitation during that time, and a closer look indeed revealed that this cannot be the reason for SWE increase. As pointed out by you and Reviewer 2 the cause most probably is a measurement error by the SSG (building and later weakening of snow bridges). We added explanation to the manuscript.*

- Finally, what was the precip for the case explained in L316-317?

  *There was no precipitation during the presented time, and hence we added the sentence: "There was no precipitation recorded in that period (last precipitation measured on December 25) and air temperatures were constantly well below 0 °C."*

- There are several situations where the pressure measured SWE is wrong. For example, see the described case in the paragraph above or the SWE increase and concurrent stable snow depth during the second half of March in Fig. 8. Please elaborate. I suggest to also check the plausibility of the calculated density as provided in 9b. The reason of the difference described in L246-247 is probably also a such wrong SWE measurement and not the difference in measurement location.

  *This issue could also be explained by building and weakening of snow bridges when the isothermal front reaches the snow-ground boundary. We also agree that this could be a problem for the calculated density in Fig. 9b. However, we are pretty*

sure that also the location of the device plays a role: please see the webcam images from June 1st and June 13th 2020 below (SPA and SSG in the background). The SSG measures 100 mm to 25 mm SWE during that time while the ground below the USH-9 (not visible here) is already snow-free. That said, we added a respective discussion and description about both these error sources to the manuscript and we also will add a remark to the data documentation in the PANGAEA repository.

[Figure]

[Figure]

Specific comments:

L34: Matiu et al. 2021

We corrected the reference year to 2021.

L55: Since the paper will not been published before summer 2021 I'd recommend to also include the winter season 2020/21.

Thank you for the suggestion. While this would be technically possible, we decided together with the data curators of PANGAEA to upload the data in annual chunks (calendar year) to keep the structure consistent, specifically for the continuation of the time series which will be uploaded to PANGAEA annually.

L63: (same special issue)?

We removed the statement, as we have no information yet if the manuscript could be added to the special issue of the preceding publication (Strasser et al. 2018).

L67: The Rofenache river

We added "river".

Fig. 1: Very bad map. Not even valleys or ridges are easy recognizable. Many geographical locations described in the text are missing in the map.

Thank you for the comment. We completely redesigned the map and added all relevant geographical locations.

L95: "..the existing weather stations..." how many? L96: "..at several locations.." What do you want to say?

We added the number of stations and changed the sentence which now reads: „ Since the reported state of the technical instruments in the catchment in Strasser et al. 2018, three of the eight existing weather stations have been extended and modified."

L113: 1.5 m does not make sense for high alpine AWS? What is the reason. Add the exact height above ground for each sensor Table 1-3. This is important for many applications. Moreover, it is in contradiction to the min/max height of 2 m written e.g. here: https://doi.pangaea.de/10.1594/PANGAEA.918096

Thank you for the comment and our apologies, this is a mistake. The instruments are all at +-2m height above the ground. 1.5 m is the height of the Pluvio constructions at the Bella Vista and Latschbloder sites which both sit on top of heavy rock boulders to be more elevated (see Fig. 2, top picture on the left and Fig. 3 in the background). We corrected the statement in the manuscript.

L123: 10 min mean values?

Depending on the variable, the logger writes 10 min. mean, max, or instantaneous values and transmits the data via GSM.

L124; I suggest to use HS instead of SD, because it is the official abbreviation.

Thanks for pointing this out, we now use HS instead of SD in the manuscript.

L126: ...by two European Avalanche...

We changed it to "… two…"

L134: Why do you mention Sommer SSG-2 and not also accordingly the same for the snow depth and snow temperature sensors?

Thank you, we removed „Sommer SSG-2" here to be consistent.

L136: The new instruments complement...

Corrected.

L142: .. installed at the main station

Added.

Fig2: The red arrow marks the main "exposed" AWSS. The blue...

Corrected.

L155: Why Sommer is mentioned for the SIR sensor, but not the SCA and the SPA-2 sensors? Be consistent!

We removed the manufacturer name "Sommer" in the text and only keep it in the Tables 1 – 3.

L163: time resolution, raw data , quality controlled?

We added the information about time resolution and quality control.

L164: I'd recommend to provide PDFs about the instruments used instead of manufacturer URLs, which can change any time.

Thank you for this suggestion! We already collected the PDFs about the instruments and will ask PANGAEA to provide them within the repository.

L171: time zone?

We added information about the time zone (UTC+1) to the manuscript.

L180: How do you manage to have enough power for heating?

The Bella Vista station has power supply from the "Schöne Aussicht Schutzhütte". At the Proviantdepot, there are four solar panels and two 72 Ah rechargeable battery packs installed. The heating device is only activated if air temperatures drop below 4 degrees C and if the signal of the USH-9 snow depth sensor is disturbed by falling precipitation. We added information to the manuscript (table captions). We also corrected an error in the manuscript text about this (Bella Vista and Proviantdepot have heated Pluviometers, Latschbloder is unheated).

L190: In 4.2.1, there is only snow depth described despite the SWE mentioned in the title.

We added a description of the presented SWE data.

Fig 7: Please provide the same figure for SWE.

We now provide the same figure for SWE in a panel plot.

L323: the technical details of the instruments are not all described!

We rephrased the sentence.

L324: It's hard to believe that no manual measurements were performed during the three years to check the plausibility and representativity of the automatic snow measurements?

We agree that manual measurements specifically for SWE would have been very desirable. However, as these stations – and particularly Proviantdepot - are only accessible via a helicopter during most of the winter time (access is limited due to avalanche risk), we did not perform them. Following your argumentation above, a series of manual SWE measurement would be necessary to get measurements of the critical moments in SWE development. Theoretically this would have been possible for Bella Vista, as this is the most accessible station, and we will consider this for the next winter season. However, there will still be no evidence for these critical cases at the other stations.

L335: Can you tell anything about funding?

Funding is a collective effort and already mentioned in the acknowledgements.

Table 1: EE08 instead of E08.

Corrected.

Is the air temperature ventilated?

Yes, air temperature is ventilated, and we added this to the manuscript.

Is the radiation sensor ventilated?

Yes, the radiation sensor is ventilated, and we added this to the manuscript.

What is the source of the given accuracy? It should rather be given in percentage.

The given accuracies are the ones provided by the sensor manufacturers as stated in the table caption.

Table4: The calculated snow drift values are wrong!

This seems to be a misunderstanding: the snow drift values were not calculated from the snow depth or SWE differences but measured by the snow drift sensor and converted to snow depth equivalents using an assumed density of 100 kg/m$^3$ (as explained in the manuscript). This was done to facilitate a comparison to the difference values for precipitation, snow depth, and SWE during that time period. The equivalent calculation was done for all measurements (italic numbers show equivalent values using the density assumption). This table, however, seems to be misleading, hence we replaced it with an explanation of the values in the text.

---

## Author Comment (AC2)

**Reply to RC2 (Author Comment on essd-2021-68)**

First of all, we would like to thank you for your thoughtful review of our manuscript and your very valuable suggestions and comments. We answer your questions and address your remarks in the following (response in blue).

Anonymous Referee #2

Referee comment on "Operational and experimental snow observation systems in the upper Rofental: data from 2017–2020" by Michael Warscher et al., Earth Syst. Sci. Data Discuss., https://doi.org/10.5194/essd-2021-68-RC2, 2021

1 General comments

Warscher et al. present a data set of non-standard measurements of snow properties together with meteorological variables at three alpine stations in the Austrian Alps.

However, the unique part (i.e. non-standard measurements at three sites) of the dataset is only available for the beginning of this year and partially for the 2019/2020 snow season. I do not see that this short period is particularly useful compared to other published multi-year datasets (e.g., Morin et al., 2012; Ménard et al., 2019). Therefore, I suggest rejecting the manuscript at this stage and waiting a few more years until more data are available.

We agree that the three years (and in some parts even less) of data are still a comparably short period of time and do not represent a long-term climatological or snow-hydrological dataset. The aim of the manuscript, however, is to follow the principles of the ESSD 'living data process' (https://www.earth-system-science-data.net/living_data_process.html) of continuous data documentation and to describe and document the state of the measurement network including the new and innovative sensor techniques. Furthermore, the newly installed sensors allow for addressing research questions intensely discussed at present (e.g., the quantification of snow drift using the innovative snow drift sensor at Station Bella Vista; this process representing the origin of massive lee-side snow loads which triggered a deathly avalanche in December 2019). Hence, our intention is to present our techniques, data and results to the scientific community as soon as possible. The station network in its current state will be maintained and further continue to record data in the described setting and we will continue to upload the data to the Pangaea repository.

In addition to the short time frame, I also agree with the first reviewer that the data quality checking is unclear and that conclusions from the data are sometimes incorrect (which I will show in the next section).

(we will address these issues below)

In addition, I would like to point out that data gaps are a major problem with this dataset. These three issues should be considered before submitting a new manuscript in a few years.

Thank you for this comment. It is true that there are gaps in the dataset. The presented stations are located at very remote and exposed high-alpine locations, two of them only accessible by helicopter during most of the winter time. We permanently do our best to maintain the systems and keep everything running, however logger failures can occur and cannot always be fixed immediately. We decided not to apply gap filling methods, but provide the raw data with only some basic error filtering. If and how the gaps should and

can be filled is depending on the application and should be decided by the respective user. Users report us that the measurements are still very valuable despite the existing data gaps.

2 Specific comments

2.1 Short time period

I can identify useful and unique parts of this data set, but in too short a time period. These are the spatial distribution (three measurements) in a high alpine environment, with non-standard snow measurements such as SWE from different gauges, liquid and solid fractions of snow, snow temperatures, and snowdrift sensors combined with standard measurements such as snow depth, precipitation, and meteorological variables. However, there is only one complete snow season (2019/20) in which more than one SWE measurement is available, but at leasot one of those is exposed to wind erosion, and the usefulness of this location will not become apparent until the start of the 2020/21 winter season, when a nearby wind-sheltered station was established. Similarly, the non-standard drift sensor and Snow Pack Analyzer (SPA) measurements are not available before early 2020. Therefore, I don't see much use for this data set described here. However, I can very well imagine one developing in a few years.

While we agree that the data presented here does not represent a long time series by now, we still think it is useful to publish the sensor and station setups including first time series now as the dataset is growing at the Pangaea repository (please see comment above). On the long run, the presented dataset will be part of a long and continuous period of snow and climate observations in the Rofental, following the 'living data process' (https://www.earth-system-science-data.net/living_data_process.html) of ESSD.

2.2 Data quality example and quality checks

Since this dataset is not standard and prone to errors, I propose to address typical measurement errors and possible automatic quality check routines in a next manuscript version.

We generally agree that automatic quality checks are useful, however, we decided to screen the data for obvious logger or sensor failures (and automatically correct them) and leave the remaining data as raw as possible (e.g., no data gap filling, no aggregation, no correction of unclear errors, etc.). This approach allows the scientific community to use and correct the data using methods fitting the respective purpose. Concerning the SWE measurements we now point to the possible errors you address below. A comprehensive comparison of the various automatic quality check routines to be applied to our Rofental data will be content of a dedicated scientific paper.

Here I would like to describe an erroneous SWE measurement that has already been identified by reviewer 1 as a misinterpretation by the authors. In lines 225ff, the authors described the stage at which the snowpack at Latschbloder becomes isothermal (Figure 8) and explained the subsequent SWE values. This is a typical time when pressure sensors measuring SWE exhibit errors (Johnson and Schaefer, 2002). I do not claim to provide the correct interpretation, but the authors certainly missed something.

This description should serve as an example of how future quality control can be designed or how errors in the data set can be described in a future manuscript, especially when more instances of redundant SWE measurements will be available (SPA and snow scale).

The authors claim snowmelt starting at midnight on 11 April 2020, explaining the loss of SWE of ~110 mm in less than 18 hours. It is questionable whether this is melt, as air temperatures were well below 0 °C and the snow depth sensor only indicated a constant decrease similar to the days before and after.

It appears to be more a measurement error which is typical when the isothermal front reaches the snow-ground boundary as described by Johnson and Schaefer (2002), which was detected based on the snow temperature measurements two days earlier. A decrease in SWE could be explained by snow shear strength being able to bridge the sensor (Johnson and Marks, 2004), which could happen when meltwater near the ground refreezes. The authors describe the later increase in SWE as rain percolating into the snow. However, reviewer 1 correctly pointed out the negative air temperatures during this time (colder than -5 °C). In addition, the rain gauge measured only <3 mm of precipitation, while the SWE sensor increases by more than 150 mm during the same period through April 18. This discrepancy cannot be explained by undercatch of the rain gauge, especially since the snow depth sensor shows the same continuous decrease as in the days before, without any indication of a major snowfall. Thus, it seems more likely that the previously mentioned snow bridges have been continuously weakened as snow temperatures are around the melting temperature. Future availability of redundant SWE data, snow depths, air and snow temperatures will provide more examples in a few years from which the authors can select examples of faulty and good situations. The methods of Johnson and Marks (2004) or others may be included to tag poor quality data.

Thank you very much for revealing our misinterpretation here and please see also our answer to Reviewer 1 regarding this issue. We fully agree with your findings and the most probable explanation for the erroneous measurements (formation and weakening of snow bridges). We removed our interpretation attempt from the manuscript and added the suggested literature (Johnson and Schaefer 2002, Johnson and Marks 2004, as well as Egli et al. 2009) to the manuscript and discuss the sources of measurement errors and the reliability of the instrument in general. We will additionally tag the data and add a respective remark to the PANGAEA repository.

2.3 Data gaps

The use of this dataset is also limited due to data gaps, which is partially visible in Figure 5. For example, for Bella Vista in 2018, over 33% of all data are missing with gap sizes of 49, 27, 20, ... days. Precipitation is missing 75% of the time, with another gap of 199 days. In 2019, this station typically has 7% data gaps, wind over 11%, with gap sizes of 12, 7, 3, <1 days. Such multi-day data gaps are difficult to fill.

Yes, there was a major problem with the logger at the Bella Vista in 2018 (and 2017). However, this holds true for Bella Vista only, not at all for the other two stations. The problem at Bella Vista station is fixed, as you can see in the data from 2019. We are not sure what you mean by the statement "typically 7%" data gaps: depending on your calculation or variable or something else? You are right, these data gaps are difficult to fill, however, unfortunately they are to be expected at hard to access alpine stations at nearly 3000 m a.s.l. (e.g., our snow pillow at Bella Vista is the highest one all over the European Alps to our knowledge). We still think these time series are very useful for many scientific applications (see also comment above on data gaps), and we are confirmed by many of our data users.

2.4 Other

- A measurement height of 1.50 m does not seem sufficient in alpine terrain. Please provide the exact height for each sensor in the tables. Please provide time periods when a sensor is buried or consider larger masts (if possible).

Unfortunately, this was a mistake (see also answer to Reviewer 1): the instruments are all at +-2m height above the ground. 1.5 m is the height of the Pluvio constructions at the Bella Vista and Latschbloder sites which both sit on top of boulders to be more elevated (see Fig. 2, top picture on the left and Fig. 3 in the background). We corrected the statement in the manuscript. The sensors were not buried within the presented time period.

- Literature describing the quality of the snow pack analyzer should be added. For example, Staehli et al. (2004) and Egli et al. (2009).

Thank you for the suggestion, we added the references to the respective section. We additionally added a section in the text about the performance of the SPA.

- The fact that each year and station is in individual files makes it difficult to use the data. Please consider consolidating the data into one or a few file(s) as much as possible.

We considered this option as well as other different consolidation and/or splitting strategies (hydrological, glaciological, etc. year). We also discussed this issue with the data curators at PANGAEA. We followed the advice of PANGAEA and decided on one file per station per year. This way we can continuously add data to PANGAEA without having to "override" the unique DOIs of a data set.

References
Egli, L., Jonas, T., and Meister, R. (2009). Comparison of different automatic methods for estimating snow water equivalent. Cold Regions Science and Technology, 57(2-3), 107-115.

Johnson, J. B., and Schaefer, G. L. (2002). The influence of thermal, hydrologic, and snow deformation mechanisms on snow water equivalent pressure sensor accuracy. Hydrological Processes, 16(18), 3529-3542.

Johnson, J. B., and Marks, D. (2004). The detection and correction of snow water equivalent pressure sensor errors. Hydrological Processes, 18(18), 3513-3525.

Ménard, C. B., Essery, R., Barr, A., Bartlett, P., Derry, J., Dumont, M., Fierz, C., Kim, H., Kontu, A., Lejeune, Y., Marks, D., Niwano, M., Raleigh, M., Wang, L., and Wever, N. (2019). Meteorological and evaluation datasets for snow modelling at 10 reference sites: description of in situ and bias-corrected reanalysis data, Earth Syst. Sci. Data, 11, 865–880, https://doi.org/10.5194/essd-11-865-2019, 2019.

Morin, S., Lejeune, Y., Lesaffre, B., Panel, J.-M., Poncet, D., David, P., and Sudul, M. (2012). An 18-yr long (1993-−2011) snow and meteorological dataset from a mid-altitude mountain site (Col de Porte, France, 1325 m alt.) for driving and evaluating snowpack models, Earth Syst. Sci. Data, 4, 13–21, https://doi.org/10.5194/essd-4-13-2012.

Stähli, M., Stacheder, M., Gustafsson, D., Schlaeger, S., Schneebeli, M., and Brandelik, A. (2004). A new in situ sensor for large-scale snow-cover monitoring. Annals of Glaciology, 38, 273-278.

---

## Author Comment (AC3)

**Reply to RC3 (Author Comment on essd-2021-68)**

Thank you for reviewing our manuscript and your very valuable comments and suggestions. We address your remarks in the following point by point (response in blue).

Anonymous Referee #3

Referee comment on "Operational and experimental snow observation systems in the upper Rofental: data from 2017–2020" by Michael Warscher et al., Earth Syst. Sci. Data Discuss., https://doi.org/10.5194/essd-2021-68-RC3, 2021

**General comments**

The authors present an extension of their previous ESSD publication that focuses on automated meteorological and snowpack observations collected in an alpine environment in Rofental, Austria. The authors followed the ESSD living data process to guide this manuscript, and accordingly nicely focus on extensions of the time series, instrumentation upgrades, and descriptions of some new instrument installations that offer additional insights into snow cover processes. I found the article easy to follow and was able to download and plot some of the data relatively easily, suggesting this data is readily accessible for future research applications. However, I did find some of the data incomplete and lacking a proper description of errors and uncertainties (see comments below).

We address the mentioned incomplete description of errors and uncertainties at the respective comments in the following.

**Specific comments**

Based on Fig. 6 it looks like some of the snow depth and SWE measurements have some missing values. Please specify why in the text.

Yes, it is true that there are missing values in snow depth and SWE measurements, mainly because of intermittent data logger failures. We added a respective explanation to the text.

Overall the text is light on descriptions of the uncertainties in the measurements, with only the instrument resolution listed in the tables. The paper would benefit from better discussion of sources of error.

Where possible, we added a more in-depth uncertainty discussion for SWE, the SPA and for the snow drift measurements. Due to the inaccessibility of the stations for longer periods during most of the winter time we were not able to do, e.g., manual measurements for comparison (please see also answer to Reviewer 1). However, we state more clearly now which measurements are potentially prone to larger uncertainties.

Some of the time series for the new sensors are relatively short in duration, such as the snow measurements at Proviantdepot. It would make sense to include data from the entire 2020-2021 winter season now that it is mostly completed.

While this would be possible, we decided together with the data curators of PANGAEA to annually upload the data (calendar year) to keep the file structure consistent, and in order to stay consistent in the continuation of the PANGAEA data time series repository.

When downloading the tab-delimited data I found it difficult to work with the column headers because the variable name, units, and method/device details were all in the same cell. If working with a scripting language like R or Python it is much easier when the columns can be indexed with short concise name, in which case the units and descriptions could be on their own rows. That being said, I am not familiar with the standards and limitations of the PANGEA data platform.

We fully agree on this. However, we are following the standards of PANGAEA in this. The advantage is that all metadata is included in the file.

**Technical comments**

Line 118: Please define the acronym "GSM"

The sentence now reads "… transmitted by means of mobile network GSM (Global System for Mobile Communications)".

Lines 155-157: Please provide more description of the SPA instrument, and how it can be used to calculate density and SWE as later shown in Fig. 9.

We now provide an in-depth description of the SPA instrument, its measurement principle and performance, and we and added the respective references.

Line 167: By "daily values" I assume this means daily averages.

Yes, with „daily values" we mean daily averages for temperature, humidity, radiation, and wind speed. To avoid misunderstandings, we changed the wording to:

"Daily averages are shown for air temperature, relative humidity, short-wave radiation, long-wave radiation, and wind speed, as well as monthly totals for precipitation."

This wording is also applied to the caption of Fig. 5.

5 and 6: These plots are missing the subplot labels (a-f)

Thank you! We added the missing subplot labels in Fig. 5. In Fig. 6 they are already included.

Fig 9: Would it be more logical to move snow depth from subplot (d) to (a) since it is the first plot discussed in the text?

We arranged the order of the subplots according to their importance, the SPA recordings hence being positioned at the very beginning.

Line 301: Can you provide an actual distance instead of "in close proximity"

We measured the distance (appr. 800 m horizontal distance) and added it to the manuscript.

Table 4: I don't understand the need for the final 2 rows in this table since they simply repeat the same values. It's also unclear why some values are in italics. Perhaps the 6 unique values presented in this table could simply be stated in the text, along with an explanation of how they relate to each other.

We intended to make the comparison clearer with the table, but it obviously creates more confusion than needed (see also comments from other reviewers). Hence, we decided to remove the table and state the values with an explanation in the text according to your suggestion. Thank you!

Fig 13: The caption description of the avalanche should use proper avalanche terminology. The edges are called 'fracture lines' with the one along the top called a 'crown' and the ones along the sides called 'flanks'.

We changed the terminology accordingly.

---

## Author Comment (AC4)

**Reply to RC4 (Author Comment on essd-2021-68)**

Thank you for reviewing our manuscript and the valuable suggestions and remarks. We address all issues and questions in the following (response in blue).

Anonymous Referee #4

Referee comment on "Operational and experimental snow observation systems in the upper Rofental: data from 2017–2020" by Michael Warscher et al., Earth Syst. Sci. Data Discuss., https://doi.org/10.5194/essd-2021-68-RC4, 2021

General comments:

The manuscript 'Operational and experimental snow observation systems in the upper Rofental: data from 2017-2020' by Warscher et al. provides a description of different types of continuously recorded snow and meteorological datasets - using standard as well as experimental sensors - collected at three sites in the Rofental in the European Alps. The manuscript is an extension of the ESSD paper 'The Rofental: a high Alpine research basin (1890–3770ma.s.l.) in the Oetztal Alps (Austria) with over 150 years of hydrometeorological and glaciological observations' by Strasser et al. 2018.

Although the title and the abstract imply that all data has been available since 2017, a closer look reveals that some datasets do not start before 2019 or even 2020. In addition, data gaps are an issue that has not been discussed in detail. I agree with Reviewer 2 that the covered time period for some recordings (especially for the unique experimental snow measurement setups) is too short for publication at current state. Therefore, I also recommend waiting some more years and collecting a longer time period of data before publication.

We here repeat the statement and our opinion on this issue that we gave to Reviewer 2: we agree that the three years (and in some parts even less) of data are a comparably short time slice and do not represent a long-term climatological period. The aim of the manuscript, however, is to follow the principles of the ESSD 'living data process' (https://www.earth-system-science-data.net/living_data_process.html) of continuous data documentation and to describe and document the state of the measurement network including the new and innovative sensor techniques. Particularly the newly installed sensors allow for addressing research questions intensely discussed at present (e.g., the quantification of snow drift using the innovative snow drift sensor at Station Bella Vista; this process representing the origin of massive lee-side snow loads which triggered a deathly avalanche in December 2019). Hence, our intention is to present our techniques, data and results to the scientific community as soon as possible. The station network in its current state will be maintained and further continue to record data in the described setting and we will continue to upload the data to the PANGAEA repository.

Concerning the data gaps, we added some more details in the text of the manuscript. Here we repeat our argumentation for Reviewer 2 as he raised the same issue: it is true that there are gaps in the dataset. The presented stations are located at very remote and exposed high-alpine locations, two of them only accessible by helicopter during most of the winter time. We permanently do our best to maintain the systems and keep everything running, however logger failures can occur which cannot always be fixed immediately. We decided not to apply gap filling methods, but provide the raw data with only some basic error filtering. If and how the gaps should and can be filled is depending on the application

and should be decided by the respective user. Users report us that the measurements are still very valuable despite the existing data gaps.

In general, I agree with the general and specific issues raised by Reviewer 1 and 2 as well as the specific/technical comments raised by Reviewer 3 and will not repeat them here again. In particular, information on assessment data quality should be included.

We included a brief description of the data quality assessment in the manuscript. The issues of the other Reviewers are addressed in the respective Author comments / answers to the reviewers.

However, I see good potential for publication in a few years (i.e. after extending the dataset for approx. two more years: 1) There is a great need for standard and experimental continuous snow monitoring datasets that cover longer periods in high-alpine regions, as such datasets are still very sparse. 2) The Rofental research catchment seems to be an ideal site for glacier, snowpack and hydrological model applications and developments, especially since the basin is not influenced by hydropower structures.

Thank you for this statement, please see our comment above on the length of the presented time series; we follow the principle of the ESSD 'living data process', our manuscript is intended to be the second publication in a continuous series.

As the authors are focusing on datasets for snow observation, it would be wise to include and describe also the other snow measurement sites in the Rofental research basin (stations Hintereisferner and Vernagtbach) in this manuscript, although they were already introduced in Strasser et al. 2018. Adding these two sites in the manuscript would make the multi-station dataset even more valuable.

We included reference to these two stations in Section 4. The respective data recordings will be published in a next joint publication.

I agree with Reviewer 3 that the data provided on the PANGEA platform was easily accessible and, except for the data gaps, was complete as described in the manuscript.

Thank you for checking the data at the PANGAEA repository!

Specific comments:

L. 2: The altitude of the research basin might be of interest for the reader; however, as you describe the data sets of specific measurement sites, the altitude of these sites would be at least as interesting to mention.

Thank you for the suggestion, we added the elevations of the specific sites to the abstract.

3: The expression 'original' (which is written twice in this line) seems strange in this context and implicates your work is somehow not original. Better change to: 'The dataset of our first study published in 2018 (https://doi.org/10.5194/essd-10-151-2018) contains... The time series presented here...'

We changed the sentence according to your suggestion.

Section 1: Please add some information on similar sites and studies (i.e. Ménard et al. 2019, https://essd.copernicus.org/articles/11/865/2019/).

Thank you for the valuable suggestion, we added these similar studies to the introduction (Lejeune et al. 2019, Ménard et al. 2019; Morin et al. 2012).

Lejeune, Y., Dumont, M., Panel, J.-M., Lafaysse, M., Lapalus, P., Le Gac, E., Lesaffre, B., and Morin, S.: 57 years (1960–2017) of snow and meteorological observations from a mid-altitude mountain site (Col de Porte, France, 1325 m of altitude), Earth Syst. Sci. Data, 11, 71–88, https://doi.org/10.5194/essd-11-71-2019, 2019.

Ménard, C. B., Essery, R., Barr, A., Bartlett, P., Derry, J., Dumont, M., Fierz, C., Kim, H., Kontu, A., Lejeune, Y., Marks, D., Niwano, M., Raleigh, M., Wang, L., and Wever, N. (2019). Meteorological and evaluation datasets for snow modelling at 10 reference sites: description of in situ and bias-corrected reanalysis data, Earth Syst. Sci. Data, 11, 865–880, https://doi.org/10.5194/essd-11-865-2019, 2019.

Morin, S., Lejeune, Y., Lesaffre, B., Panel, J.-M., Poncet, D., David, P., and Sudul, M. (2012). An 18-yr long (1993-–2011) snow and meteorological dataset from a mid-altitude mountain site (Col de Porte, France, 1325 m alt.) for driving and evaluating snowpack models, Earth Syst. Sci. Data, 4, 13–21, https://doi.org/10.5194/essd-4-13-2012.

58-60: As you are describing snow drift measurements in detail (Section 4.2.4), I would recommend to introduce this point already here, i.e. extending point I to : I) Improved process understanding of snow drift, accumulation and melt dynamics in high mountain regions.

We changed the sentence according to your suggestion.

92-93: Information on topography and meteorological conditions of the research site should be moved to Section 2.

Yes, we fully agree. We moved the information to Section 2.

Section 3.1 and 3.2: Several statements (especially the site descriptions, coordinates) are repetitive. I would suggest merging these two subsections and describing each site individually introducing their meteorological and snow sensors together in one subsection.

That is true, we followed your suggestion and merged the two sections.

Section 4.2.4: This section is very long compared to the other subsections of 4.2. I would suggest to describe the snow drift measurements in general in this section and move the explicit case study to a new section (i.e. Section 5: Case study - Application of the dataset for an improved assessment of avalanche-critical blowing snow situations).

Thank you for the suggestion! We included a new section for the avalanche case study.